# Epidural and Intrathecal Drug Delivery in Rats and Mice for Experimental Research: Fundamental Concepts, Techniques, Precaution, and Application

**DOI:** 10.3390/biomedicines11051413

**Published:** 2023-05-10

**Authors:** Md. Mahbubur Rahman, Ji Yeon Lee, Yong Ho Kim, Chul-Kyu Park

**Affiliations:** 1Gachon Pain Center and Department of Physiology, Gachon University College of Medicine, Incheon 21999, Republic of Korea; mahbub84@gachon.ac.kr; 2Department of Anesthesiology and Pain Medicine, Gachon University, Gil Medical Center, Incheon 21565, Republic of Korea; easy956@gilhospital.com

**Keywords:** epidural route, intrathecal route, drug delivery, rats, mice

## Abstract

Epidural and intrathecal routes are the most effective drug administration methods for pain management in clinical and experimental medicine to achieve quick results, reduce required drug dosages, and overcome the adverse effects associated with the oral and parenteral routes. Beyond pain management with analgesics, the intrathecal route is more widely used for stem cell therapy, gene therapy, insulin delivery, protein therapy, and drug therapy with agonist, antagonist, or antibiotic drugs in experimental medicine. However, clear information regarding intrathecal and epidural drug delivery in rats and mice is lacking, despite differences from human medicine in terms of anatomical space and proximity to the route of entry. In this study, we discussed and compared the anatomical locations of the epidural and intrathecal spaces, cerebrospinal fluid volume, dorsal root ganglion, techniques and challenges of epidural and intrathecal injections, dosage and volume of drugs, needle and catheter sizes, and the purpose and applications of these two routes in different disease models in rats and mice. We also described intrathecal injection in relation to the dorsal root ganglion. The accumulated information about the epidural and intrathecal delivery routes could contribute to better safety, quality, and reliability in experimental research.

## 1. Introduction

Delivering a therapeutic agent to the central nervous system (CNS) is challenging because of the blood–brain barrier and blood–spinal cord barrier of the CNS, which consist of endothelial cells possessing tight junctions and efflux pumps [1]. Thus, the administration of drugs via parenteral routes ultimately results in low concentrations of drugs in the CNS because of the blood–brain and blood–spinal cord barriers. To overcome these par-enteral delivery challenges, intraspinal drug delivery systems are one of the most common approaches used. The epidural [2,3,4] and intrathecal routes [5,6,7] are the most used intraspinal drug delivery routes in rats and mice. The target of each drug delivery route is the cerebrospinal fluid (CSF). With an intrathecal injection, a drug is delivered directly into the CSF within the intrathecal space of the spinal column [1]. Intrathecally administered drugs are restricted within the CSF circulating in the spinal column and brain ventricles. However, drugs delivered to the epidural space cross the dura membrane to reach the CSF. Thus, epidurally administered drugs can also reach systemic circulation. These two routes are compared in Table 1.

Intraspinal drug delivery through an epidural or intrathecal injection is comparatively more challenging in small experimental animals such as rats and mice than in large animals and humans because of the small size of the rat and mouse spinal column, spinal canal, epidural space, and intrathecal or subarachnoid space, and the small volume of CSF. Substantial experimental research and a review of the literature in human medicine have been reported [1,8,9,10]. Numerous research studies have used the epidural [4,11,12] and intrathecal routes [13,14,15] in mice and rats; however, to the best of our knowledge, no review of the literature has discussed the delivery process, challenges, and application of these routes in rats and mice.

Furthermore, the intraspinal intrathecal space [16,17] and intracerebroventricular route through the brain parenchyma into the lateral ventricles of the brain have been used to directly access the CSF [1]. However, the intraspinal/intrathecal route is broadly used because it is less invasive and is suitable for chronic use with surgical catheterization [13,18] and for direct single-dose injections via needle puncturing [6,17,19].

The objectives of this comparative study were to discuss the anatomical locations of the epidural and intrathecal spaces, CSF volume, techniques and challenges of epidural and intrathecal injection, dosage and volume of drugs, and the purpose and applications of these two routes in different disease models in rats and mice. We further describe the sites of injection, needle and catheter types and size, injection volume, concentration, and purpose and indications of epidural and intrathecal injections in experimental research in mice and rats, as well as how intrathecal injection can be effective for therapeutic agents targeting the dorsal root ganglion (DRG).

## 2. Anatomy and Physiology of the Spinal Meninges

### 2.1. Organs, Tissues from Skin to Spinal Cord

The intrathecal or epidural administration of a therapeutic agent requires an in-depth knowledge of the structures from the skin to the spinal cord and the vertebral column anatomy, as well as considerable technical skill. For intraspinal drug delivery, the most important structure is the meninges, whose function is to hold and maintain the CSF surrounding the brain and spinal cord, thereby protecting and regulating their function. The meninges are the outer coverings of the brain and spinal cord, called the cranial meninges and spinal meninges, respectively, which separate the brain and spinal cord from the adjacent bony structures of the skull and vertebral column, respectively. The spinal and cranial meninges are continuous with each other in that the unique structure of both consists of the same three meningeal layers (from superficial to deep): the dura mater, arachnoid mater, and pia mater (Figure 1 and Figure 2). In addition, the upper parts of the spinal canal in mice, rats, and other tetrapods are formed by the bony part, vertebral arch (inside a single vertebra), and ligament flavum (i.e., intervertebral space; Figure 3). The ligament flavum is important for intraspinal drug delivery, especially for epidural and intrathecal injections. The structures from the superficial to deep regions are the skin, subcutaneous fat, muscle, supraspinous ligament, interspinous ligament, ligament flavum, epidural space, spinal meninges (dura mater, arachnoid mater, and pia mater), and spinal cord (Figure 3).

### 2.2. Epidural Space

The term “epidural” is composed of the prefix “epi-”, which means “up” or “above”, and “dural”, which refers to the dura mater layer of the meninges. Thus, an epidural is the upper part of the dura mater. The epidural space is the space between the outer layer of the meninges (i.e., dura mater) and the inner periosteum or ligament flavum of the vertebral column [20]. Mice, rats, and other tetrapods have two adjacent upper parts of the epidural space: the first is the bony part (i.e., the vertebral arch (inside a single vertebra)), and the second is the connective tissue part (i.e., ligament flavum (intervertebral space)). It is filled with loose connective tissue, epidural fats, veins, arteries, internal vertebral venous plexuses, spinal nerve roots, and lymphatics [21] (Figure 1, Figure 2 and Figure 3).

### 2.3. Dura mater

The dura mater is the outermost of the three layers of the meninges and lies between the epidural space and the arachnoid mater. It consists of irregularly oriented collagen fiber bundles that accommodate blood vessels and through which nerve fibers and nerve fascicles pass [22].

### 2.4. Subdural Space

The term “subdural” is composed of the prefix “sub-”, which means below”, and “dural”, which refers to the dura mater layer of the meninges. Thus, the subdural space is the lower part of the dura mater or the space between the inner layer of the dura mater and the upper surface of the arachnoid mater. This very thin layer contains a thin film of fluid. The application of drugs to the subdural space in human medicine has been reported [23], but this application in experimental medicine in mice and rats is difficult owing to the layer’s negligible thickness. However, this space is important for experimental cranial subdural hemorrhage [24].

### 2.5. Arachnoid Mater

The origin of the word “arachnoid” is the Greek word “Arachne”, meaning “spider”, because of the web-like collagenous fibroblast structure of the arachnoid membranes, trabeculae, and septae that cross the subarachnoid space from the arachnoid mater to the pia mater [25]. The arachnoid mater underlying the dura mater comprises the outermost arachnoid barrier cell layer, which includes tightly packed cells, numerous tight junctions, and no extracellular collagen but an innermost collagenous portion. Owing to its numerous tight junctions, this layer is a physiological and morphological barrier between the subarachnoid CSF and blood circulation in the dura [26]. This layer has two parts: the basal arachnoid mater and the interweaving arachnoid trabeculae. The arachnoid trabeculae are rich in mitochondria and originate from the inner surface of the arachnoid mater to the pia mater, thereby contributing to the subarachnoid space, which contains CSF [26]. In addition, the arachnoid matter protrudes into or through the dura mater into the epidural space. This protrusion is defined as arachnoid villi or granulations (Figure 2), and it produces CSF. The vascular and lymphatic interface area has a role in CSF drainage [27]. Most arachnoid villi originate near the nerve root area and provide expansions, such as the lumbar cistern, in the lower lumbar spinal cord [28].

### 2.6. Subarachnoid Space/Intrathecal Space

As with the word “subdural”, the term “subarachnoid” is composed of the prefix “sub-”, meaning “below”, and “arachnoid”, referring to the arachnoid mater layer of the meninges. Thus, the subarachnoid space is the space between the arachnoid and pia mater, and it is more commonly called the “intrathecal space.” The prefix “intra-” means “inside”, and “thecal”, which originated from the Greek word theke, means “a hard outer covering” and refers to the arachnoid mater. The term “intrathecal” literally means “in-side the arachnoid mater”. This space contains CSF, numerous arachnoid trabeculae, col-lagen fibrils, and major blood vessels (i.e., veins and arteries) that cross the CNS to the epidural space. The subarachnoid spaces of the cranium and vertebral column are connected to each other and form a single closed route for CSF circulation [27,28]. This space is mostly used for intraspinal drug delivery in mice and rats via intrathecal injection and catheterization.

### 2.7. Pia Mater

The pia mater is the innermost layer of the meninges that directly envelops the brain and spinal cord. The pia mater consists of flattened thin fibroblast tissues, collagen micro-fibrils, and capillaries containing a vascular plexus for the spinal cord tissue. This layer is separated from the neural tissue of the spinal cord by loose collagen fibers and the glia limitans. The single-cell layer thickness and fenestration of the basal lamina in many areas of the pia expose the glia limitans directly to the subarachnoid space. Pial and arachnoid cells also envelop the nerves, collagen bundles, and blood vessels that are within or cross the subarachnoid space and those around arteries that penetrate the CNS [29]. The spinal pia mater also has ligamentous lateral projections, called denticulate ligaments, to both sides of the spinal cord. These emerge through the arachnoid and attach to the spinal dura mater. Each pair of denticulate ligaments is located between pairs of spinal nerves, and they fix and hold the spinal cord in place [30].

### 2.8. Cerebrospinal Fluid Production, Volume, and Circulation

#### 2.8.1. CSF Production and Volume

Clear knowledge about CSF production, circulation, absorption, and total volume is necessary before intraspinal drug delivery because an extra volume of any drug may change the volume and pressure of the CSF [27; 31]. CSF is a colorless fluid that is secreted by the choroid plexus into the ventricular cavities (80%) and by other structures, such as the brain parenchyma (20%) [31]. Compared to blood plasma, CSF contains a higher concentration of sodium, chloride, and magnesium ions and a lower concentration of glucose; proteins; amino acids; uric acid; and calcium, phosphate, and potassium ions [32]. The rate of CSF production and the total CSF volume vary across species (Table 2), but CSF circulates similarly inside the ventricular system and the cranial and spinal subarachnoid spaces. CSF lubricates, protects, and serves as a natural cushion for the brain and spinal cord, constantly providing ions, nutrients, hormones, and neurotransmitters and removing metabolic waste products.

#### 2.8.2. CSF Circulation

In brief, after CSF is produced by the choroid plexus of the lateral ventricles and the parenchyma of the brain, it moves to the third ventricle and then to the fourth ventricle. From the fourth ventricle, CSF enters the subarachnoid space, basal cisterns, and the cisterna magna. In the cisterna magna, CSF flows along three routes: anteriorly to the pre-medullary prepontine cisterns and cerebellopontine cisterns, superiorly to the hemispheric subarachnoid space, and inferiorly to the spinal subarachnoid space [25,31,33]. It circulates through the single, closed cranial, and spinal subarachnoid space. Finally, the CSF is absorbed into the dural venous sinuses by the cranial and spinal subarachnoid villi and granulations via diffusion, and osmotic pressure plays a role in the filtration process. The circulation of CSF is believed to occur by several actions, such as continuous production and absorption of CSF by ciliary action applied by the ventricular ependyma for different pressure gradients of the arachnoid villi. CSF circulation is also influenced by brain movement, the cardiac cycle, and arterial blood flow [25,34].

## 3. Dorsal Root Ganglion (DRG)

In general, there are 31 pairs of DRGs in mice and rats on both sides, including 8 cervical, 13 thoracic, 6 lumbar, and 4 sacral pairs [35]. DRG refers to the dorsal root of the spinal nerve, which emerges through the intervertebral neural foramen and expands to form a ganglion, called the DRG. The DRG is close to the spinal cord, located just before the dorsal and ventral spinal roots unify to form a single spinal nerve. The DRG is composed of sensory neurons, satellite glia, endothelial cells, and macrophages [36]. It is highly vascularized and contains capillary tissue supplied by the branches of the segmental arteries. In particular, the DRG has high blood vessel permeability for unique endothelial cells, allowing the biodistribution and absorption of systemic drugs, which are limited in other parts of the nervous system [36]. This characteristic of the DRG has clinical and preclinical significance for systemic coverage. However, the systemic approach is not typical because it exposes multiple tissues often related to serious adverse effects. Intra-DRG injection is widely used in humans and large animals, but it is challenging to administer in rats and mice because of their small size. The intrathecal injection is frequently used by targeting the DRG, although controversy exists regarding whether the DRG has direct contact with the CSF. An interesting study in human cadavers demonstrated that an intra-DRG injection of green ink frequently reached the spinal cord [37], indicating the DRG’s connection with the CSF and spinal cord. Intra-DRG injection of toluidine blue also consistently stains the ipsilateral CSF and spinal cord in rats [38]. Chang et al. [36] also found that an intraspinal nerve injection stained the ipsilateral spinal cord, thereby indicating that nerve tissue can convey drugs even if it has no direct contact with the CSF. Furthermore, Chang et al. [38] also demonstrated that an intrathecal injection stained the DRG and the spinal nerve. Another study [39] reconfirmed that intrathecal injection of methylene blue in rats also stained the DRG and spinal nerves. CSF and subarachnoid space reaching half of the DRG length has also been demonstrated in rats [39] and in some length of the DRG in humans [40]. Therefore, the intrathecal route can be a good choice for targeting the DRG in rats and mice.

## 4. Epidural and Intrathecal Injection Procedures

Epidural and intrathecal injections can be performed via acute needle puncture or catheterization. Acute needle puncture is usually used for single drug administration but can also be used for multiple doses [5], and intervals of 24 h, 2 days, and 7 days have been reported (Table 3). Acute needle puncture can be performed in anesthetized [41] or unanesthetized animals [5,42]; however, intrathecal injections in unanesthetized animals require greater skills. The intervertebral space of the spinal column is commonly used, but the lumbar region [41,43], especially L5–L6, is most commonly used for single acute injection [16,44]. However, multiple injections were performed through a single location, e.g., L5-L6 [5,6], L4-L5 [45], and top of the foramen [46] (Table 3). Proper needle insertion to the appropriate location and an accurate volume and concentration of the injectate, based on the species, are critical for the success and the recovery of animals; these are briefly discussed in Table 3, Table 4, Table 5 and Table 6. For an effective IT injection via acute puncture, only one try is required; however, if the first effort fails, the needle should be removed, and a second attempt may be made. If the second effort fails, a different intervertebral space should be chosen [15]. In addition, shaving and aseptic preparation of the skin, needles, and administering agents are also important for success.

The exact positioning of a needle to the epidural space is difficult noninvasively and requires high skill because the distance between the epidural and intrathecal spaces is very small in rats and mice. Therefore, the applications of direct epidural injections using needle puncture in mice and rats are limited because the needle can easily penetrate the dura mater and enter the intrathecal space. Therefore, intrathecal injection by acute needle puncture [5,6,17], intrathecal catheterization [11,13,14], and epidural catheterization [2,3,4] but not epidural injection by needle puncture is frequently used in mice and rats.

### 4.1. Procedure of Intrathecal Injection by Acute Needle Puncturing

After administering anesthesia (e.g., volatile isoflurane), the thoracolumbar region is shaved. Needles are inserted through the intervertebral space where bony parts are absent, which is covered by the ligamentum flavum [20]. The angle of needle insertion varies but has been reported at over 20° [42,47,48,49], 45° [50,51], at 45° shown in videos but reported to be 70–80° in the discussion section [16], and at 70–80° shown in videos [17] in mice; an angle of 15–30° has been reported in rats [51]. Therefore, it is difficult to conclude what the optimal injection angle is (Figure 3 and Figure 4). However, the authors of the current review also insert needles at angles of 70–80° and then reduce the angle to 30–45° during drug injection to easily spread out the injected drug from a narrow needle and to prevent CSF leakage during needle withdrawal (Figure 4). During insertion, when the tip of the needle reaches the bottom of the spinal canal (upper part of the vertebral body), the needle should not be pushed further as it may penetrate the intervertebral disc into the abdominal cavity [52] (Figure 1 and Figure 3). The depth of needle insertion has been reported to be approximately 0.30 cm in rats after skin and muscle incision and 0.30–0.40 cm without skin and muscle incision in mice [50].

The iliac crest and the L5–L6 intervertebral space are located by detection of the supraspinous process (Figure 1) of these two vertebrae (Figure 5). The pelvic girdle is then softly held by one hand to fix the dorsoventral position, and, on the other hand, a sterile needle is inserted at the appropriate angle to penetrate the ligamentum flavum and dura mater to reach the arachnoid space. However, some researchers use 1–3 cm longitudinal skin incisions over the spinous process of the desired intervertebral space in rats for better confirmation of the location during a single acute intrathecal needle puncture [51,53,54]. The drug is then delivered slowly to allow for spreading. The delivery time varies (Table 3) and should be selected considering the characteristics of the injected solution and the species. In addition, after drug delivery, care should be taken to wait 15 s to 1 min before withdrawing the syringe (Table 3); otherwise, drugs may be pulled back into the syringe. Appropriate intrathecal positioning can typically be confirmed via the tail-flick test, but this response does not occur every time. Dura mater puncture can also be confirmed by other characteristics, such as the formation of an “S” shape by the tail, by hind paw retraction, and occasionally by backflow of the CSF [14,15,20]. After injecting a drug, temporary motor paralysis also occurs, which is a sign of successful drug administration.

The injection volume is usually 5–10 μL in mice and 10–50 μL in rats, but different volumes have also been used (Table 3, Table 4, Table 5 and Table 6). An intrathecal injection of 30–50% of the volume of the total CSF volume is defined as a larger volume [27]. After the thoracolumbar intrathecal administration of a larger volume, it is immediately transferred to the cere-bro-cervical region. Thirty-three and forty-two percent of the total CSF volume in a thoraco-lumbar intrathecal administration was reported as tolerable in humans and non-human primates, respectively [27]. However, the limits of safe injection volumes have not been characterized in mice and rats, and further experimental studies are needed. The total volume of CSF in mice and rats is 35–40 μL and 150 μL, respectively [1]. The needle size for intrathecal injection by needle puncture in rats and mice varies between articles, but a 30-G needle is commonly used for mice [5,7], and 25-G is used for rats [15,17] (Table 3 and Table 4).

Furthermore, the dead space is sometimes ignored, but it has a significant impact on the accurate dosing of a drug because a very small total drug volume is injected via these routes. Dead space is the internal volume of the catheter or needle through which a drug passes from the syringe to the targeted region (e.g., epidural or intrathecal space). If the dead space is 10 µL and the injected volume is 10 µL, then the entire amount of the drug will remain in the catheter. To solve this problem, the dead space can first be filled with the injected agent, and the targeted volume should then be withdrawn in the syringe and then injected. Another method is by flushing with an equal volume of saline to replace the dead space occupied by the drugs [3,14,18]. However, care should be taken to prevent air bubbles when changing the syringe, which may cause adverse effects such as the alteration of subarachnoid pressure and injury to the nerves or meninges [55].

### 4.2. Epidural and Intrathecal Catheterization

Epidural and intrathecal catheterization are convenient and less stressful for animals when applying multiple doses for long-term medication. Intrathecal catheterization is administered in the atlanto-occipital and thoracolumbar regions [18,43], but epidural catheterization is only administered in the thoracolumbar region (Table 4 and Table 5). Catheters are inserted into the epidural and intrathecal space either through the intraspinal space by penetrating/cutting the ligament flavum [20,43] or by laminectomy (e.g., dorsal laminectomy, lateral laminectomy) [20]. In addition, atlanto-occipital catheterization is conducted by penetrating the anterior atlanto-occipital membrane that joins the upper cranial border of the anterior arch of the atlas (C1) to the anterior inferior surface of the foramen magnum [13,18]. The catheter is usually extended from L4 to L6 until the lumbar enlargement region [11,56] because the spinal nerves of these regions have pain and motor function-related clinical significance for supplying the hind legs. However, in a spinal cord injury model, the target is to reach the injured area (e.g., from the atlanto-occipital ar-ea to the T10–T11 injured area) [57]. The length of the spinal column of rats and mice should be known before catheterization, and the length from the atlanto-occipital space to the L6 vertebra is approximately 11 cm in rats and 4.2 cm in mice (Figure 5). One study [58] showed that the length was consistently 11 cm in rats. Therefore, in atlanto-occipital catheterization, a catheter should be extended 8–11 cm in rats and 3–4 cm in mice (Table 4 and Table 5). Therefore, care should be taken to avoid dura mater puncture in epidural catheterization and to avoid spinal cord injury in intrathecal catheterization. Usually, the length of catheter insertion for thoraco-lumbar catheterization in mice and rats is 1–2 cm but varies, with a maximum reported length of 4 cm [59]. The other end of the catheter is tunneled subcutaneously, and the opening is at the skin of the neck region, and the opening of the catheter is sealed or joined to a pump. Dura mater puncture can be confirmed based on behavioral responses such as those that occur with intrathecal injection. If animals exhibit any neurological deficits arising from the surgical procedure, they should be excluded from further experiments [11]. Additionally, 3 days after surgery, 10 μL of 2% lidocaine is often added to observe temporary motor paralysis and sensory loss for 20–40 min. If this does not occur, the animal should be excluded [14,18,49,49]. For epidural catheterization, animals should be excluded if CSF is aspirated or behaviors associated with dura mater puncture are observed [2] because epidural catheters should remain in the epidural space and above the dura mater where CSF is absent. Invasive epidural and intrathecal catheterization can be performed by direct observation of the dura mater by placing the catheter above the dura mater for epidural catheterization and by puncturing the dura and maintaining the catheter below the dura mater. Dead space should also be cautiously considered because of long catheters. PE-10 tubing is commonly used for intrathecal and epidural catheterization in rats and mice (Table 4 and Table 5). There are no studies showing how long catheters can be maintained after implantation. However, accurate placement of catheters was confirmed after 45 days in mice [60] and 9 months in rats [58]. Nonetheless, after experiments, the proper placement of catheters must be confirmed [49,60,61]. The injection site, injectate volume, concentration, frequency, duration, and purpose of epidural and intrathecal injections by catheterization are discussed in Table 3, Table 4, Table 5 and Table 6.

**Table 3 biomedicines-11-01413-t003:** Intrathecal injection by acute needle puncture in mice and rats.

Species	Injection Site	Volume	Syringe Size	Time of Injection	Time of Syringe Withdrawal	Number of Injections	Reference
C57BL/6 mice	L4–5	10 μL	30-G needle to 50-μL Hamilton	3 μL/min	The needle was removed 1 min after completion and was kept in Trendelenburg position 5 min more	Single	[17]
C57BL/6 mice	L5–6	10 μL	30-G 0.5-in needle	-	-	Three injections, 24-h intervals	[7]
Kunming mice	L5–L6	10 μL		Delivered for more than 30 s	Syringe maintained for an additional 15 s to ensure diffusion before removal	Single	[6]
C57BL/6J mice	L5–6	5 μL	30-G in 10-μL Hamilton	-		Three injections at two-day intervals	[5]
FVB/NJ mice	L5–L6	8 μL	27-G needle 25-μL Hamilton syringe	1 μL/4 s	1 min after finishing delivery	Single	[16]
Mice	L5–L6	5 μL	30 G	1 μL/6 s	15 s	Single	[47]
C57BL/6 mice	top of the foramen magnum	20 μL	25-G, 1-mL syringe	Slowly	After 2 min	Three injections at 7 days intervals	[46]
CDI mice	L5–6	10 μL	30-G needle	-	-	Single	[62]
Mice		20 μL	30-G 1/2 in 50 μL Hamilton	Injections were delivered as a bolus within 5 s		Single	[63]
SD rat	L2–3	0.2 mL or 2 mL	1-mL syringe	1-mL syringe	After injection, rats placed upside-down at a 45° angle for 15 min	Single	[54]
SD rat	L5–6	30 μL	31 G		-	Single	[44]
Wistar rats	L4–5	15 μL	26 G	3 μL/min	-	Two injections, 24-h intervals	[45]
Wistar rats	L6–S1	0.02 mL/kg, average of 0.05 mL per rat	25 G	1 mL/min, average: 3 s/injection	1 mL/min, average: 3 s /injection	Single	[15]
Wistar rats	L3–4 or L4–5		25 G	1 min		Single	[53]

**Table 4 biomedicines-11-01413-t004:** Intrathecal injection by catheterization in rat and in mouse.

Species	Site of Insertion	Catheter Size and Total Length	Inserted Length	Dead Space and Filling Agent	Injected Volume	Reference
Lumbar						
SD rat	L4–5	PE-10 (0.6 mm diameter)	1–2 cm	20 µL, saline	10 μL	[11]
SD rats	L4–5	PE-10 tube, 12 cm	2 cm	-	-	[43]
SD rats	L5–6	PE-10 (0.6 mm diameter), 10 cm	4 cm	-	10 μL	[59]
SD rats	L5–6	PE-10 (0.6 mm diameter), 15 cm	3 cm	4.5 µL, saline (7 µL)	10 μL	[14]
SD rat	L2 laminectomy, tip located between L3 and L5	SUBL-14	L3–L5	10 µL	25 or 50 μL	[64]
Rats	T13–L1	PE-5 catheters (outside diameter: 0.36 mm)	L2–L5	6 µL, PBS	20 µL	[60]
Atlanto-occipital						
SD rat	Atlanto-occipital	ALZET catheter (PU-10 28G	8 cm caudally to reach lumbar enlargement	10 μL, sterile saline	20 μL	[18]
SD rat	Atlanto-occipital	PE-10	8.5 cm caudally to reach lumbar enlargement	10 μL, sterile saline	10 μL	[65]
Mice	Atlanto-occipital	-ALZET IT mice catheter -O’Buckley IT catheter	2.5 cm			[13]

**Table 5 biomedicines-11-01413-t005:** Epidural catheterization in rat and in mouse.

Species	Site of Insertion	Catheter Size and Total Length	Inserted Length	Dead Space and Filling Agent	Injected Volume	Reference
SD rat	L4–5	PE-10 (0.6 mm diameter)	1–2 cm	20 µL, saline	10 μL	[11]
SD rat	T13–L1	PE-10	~3.0 cm until L5–6	100 µL of hyaluronic acid, 0.9% saline	100 µL of hyaluronic acid, 0.9% saline	[2]
SD art	T13–L1	PE-10 catheter	~3.0 cm until L5–6	10 µL of sa	160 µL	[3]
Mice	T11–T12	PU-10catheter	1 cm	-	50 µL	[4]

## 5. Uses and Application of Epidural and Intrathecal Injection

The epidural route is widely used for the induction of anesthesia in large animals and humans [66]. However, this route is widely used for analgesic purposes and not for anesthesia. In rats and mice, the intraperitoneal, intravenous, and intrarespiratory routes (i.e., for volatile anesthetics) are used to induce anesthesia. These routes are most commonly used for pain management with analgesics [65]. However, beyond pain management, the intrathecal route is widely used for stem cell therapy [45,46,62,67,68], gene therapy [7,17,38,68], delivery of immune cells [63], sedation [41], protein therapy, insulin delivery [69], mineral delivery [70], chemical delivery [2,70,71,72], and drug therapy using agonists [43,72,73], antagonists [43,64], antibiotics [70,74], and antiparasitic drugs [75] (Table 6). Intraspinal injection used in pain models can be divided into cancer pain models [68], including chemotherapy, induced pain models [5,75], and non-cancer pain models, such as models of arthritis [24], rheumatoid arthritis [72], diabetes-induced neuropathic pain [69,73], chronic pancreatitis-induced pain [19], spinal injury-induced pain [57,60], post-herpetic neuralgia [43], foraminal stenosis-induced pain [2,3], chronic DRG compression-induced pain [45], spared nerve injury [76], intrathecal capsaicin-induced spontaneous pain [71], chronic post-ischemia neuropathic pain [77], spinal nerve ligation-induced pain [11,44], and the acetic acid-induced writhing test [70] (Table 6). Furthermore, the intraspinal route is also used for the evaluation of safety and analgesic effects in normal healthy animals [12,65]. Beyond pain management, the intrathecal route is also used for drug administration for the amelioration of spinal injury-induced spasticity [60] and the induction of itching and scratching in behavioral models [78] and a pruritis model [42] (Table 6).

**Table 6 biomedicines-11-01413-t006:** Different uses and applications of epidural and intrathecal injections.

Species	Method of Drugs Administration	Disease Model	Types of Agents Injected	Purpose of Injection	Concentration	Injected Volume and Vehicle	Reference
SD rat	ITc	Resiniferatoxin-induced postherpetic neuralgia	-Amiloride, a potent ASIC3 inhibitor -7,8-DHF, TrkB agonist, 3 mg/kg	-To evaluate involvement of ASIC3 and TrkB signaling in pain in dorsal root ganglia	100-μg amiloride daily for 7 days -3-mg/kg, TrkB agonist for 7 days	10 μL	[43]
SD rat	ITc	Spinal nerve ligation-induced pain model	Phosphodiesterase 4B-specific siRNA	-To reduce neuroinflammation	2 μg		[11]
SD rat	ITinj	Chronic pancreatis model	Cognate receptor C–X–C chemokine receptor type 4 (CXCR4) inhibitor	-to reduce pancreatic pain	5 μg/10 μL daily for one week	10 μL	[19]
SD rat	ITc	Freund’s complete adjuvant-induced rheumatoid arthritis	Crocin	-To reduce rheumatoid arthritis-induced pain	100 mg/kg	20 μL	[72]
SD rat	ITc	Bone cancer pain model	Genetically engineered human bone marrow stem cells	-To reduce bone cancer pain	6 × 10^6^ cells	10 μL	[68]
SD rat	ITinj	Neuropathic pain	Adipose tissue-derived stem cells (ASCs)	-To relieve neuropathic pain	1 × 10^6^ cells	30 μL DMEM	[44]
SD rat	EDc	Foraminal stenosis-induced pain	Hyaluronic acid (HA)	-To relieve neuropathic pain	100 µL of HA	100 µL of HA	[2]
SD rat	EDc	Healthy rats	Gabapentin	-To evaluate safety and toxicity	30 mg	300 μL	[12]
SD rat	EDc	Lumbar foraminal stenosis-induced pain	Polydeoxyribonucleotide	-To evaluate analgesic effect	0.1 mg/kg	160 µL	[3]
Wistar Rat	ITc	Spinal cord ischemia	human umbilical cord blood stem cells	To improve spinal cord function	1 × 10^4^ HUCBSCs	10 μL	[67]
SD rat	ITc	Spinal cord injury model	Embryonic Stem Cell-Derived Spinal GABAergic Neural Precursor Cells	To reduce central neuropathic pain and motor function	1 × 10^6^ cells	-	[57]
Wistar rat	ITinj	Chronic DRG compression-induced pain model	Bone marrow stromal cell	-To reduce neuropathic Pain	1 × 10^6^ cells	15 μL	[45]
CD1 mice	ITinj	CCI-induced neuropathic pain model	Bone marrow stromal cell	-To reduce neuropathic pain	1 or 2.5 × 10^5^ cells	10 μL	[62]
Rat	ITc	Spinal cord injury-induced spasticity	-Potassium-chloride cotransporter KCC2 - BDNF	-To evaluate the involvement of KCC2 and BDNF in spasticity	20 μg 10 μg	20 μL	[60]
Rat	ITc	Phasic andincisional pain	Gentamycin, Streptomycin, Neomycin	-To evaluate Antinociceptive potency of aminoglycoside antibiotics	5 μg, 15 μg, 15 μg, respectively	10 μL	[74]
Rat	ITc	Diabetes-induced neuropathic pain	Insulin	-To evaluate Antinociceptive potency of insulin	0.2 U	10 μL	[69]
Rats, Mice	ITc	Diabetes-induced neuropathic pain	Sirtuin 1 agonist, SRT1720	-To reduce neuropathic pain	0.8 μg in rats, 1.4 μg in mice	10 μL	[73]
Rat	ITc	Spared nerve injury (SNI)	TMEM16A, U0126 inhibitors	-To find out the mechanisms of neuropathic pain	10 μg, 10 μg	10 μL	[76]
Mice	ITinj	Chemotherapy (Paclitaxel)-induced neuropathic pain	-Artesunate	-To reduce chemotherapy-induced neuropathic pain	100 μg	5 μL	[75]
Mice		Neuropathic Pain	Decursin	-To reduce pain			[5]
Mice	ITinj	Spontaneous pain	Capsaicin	To induce spontaneous pain	0.5 µg in 10 μL		[71]
Mice	ITinj	PAR-2 activator trypsin-induced scratching behavior	-gastrin-releasing peptide (GRP) -Opioids	-To reduce scratching behavior	1 nmol/5 μL	5 μL	[78]
Mice	ITinj	Morphine-induced pruritis	Morphine	-To induce scratching behavior	0.3 nmol	5 μL	[42]
Mice	ITinj	Chronic post-ischemia neuropathic pain model	Human mesenchymal stem cells	-To reduce pain behavior	2 × 10^5^ cells	5 μL	[77]
Mice	ITinj	Collagen-induced arthritis	ERK1/2 inhibitor (U0126), Tramadol, and NMDAR antagonist D-2-amino-5-phosphonovaleric acid	-To reduce pain behavior	1.6 µg, 250 µg, 0.5 µg, respectively	5 μL	[24]
Mice	ITc	Acetic acid-induced writhing test	Neomycin, gentamicin	-To evaluate antinociceptive effects	0.5–20.0 µg, 5–40 µg, respectively	10 μL	[70]

ITc, intrathecal catheterization; ITInj, intrathecal injection; EDc epidural catheterization, TMEM16A, transmembrane protein 16A.

## 6. Limitation

This review described the anatomy and physiology of spinal meninges, CSF, and DRG, focusing on epidural and intrathecal drug delivery. However, we did not include comparative studies of rats/mice with humans. The spine [79,80], meninges [81], and DRG [82] of humans and rodents have similar structures and components. Similarly, as compared to humans, rodents have a lower relative amount of CSF and slower rates of CSF circulation, but they have similar systems for production and circulation [1,31,33,81]. Therefore, we provided a general discussion of the spinal meninges, CSF, and DRG to provide clear information for researchers planning to deliver drugs epidurally and intraspinally.

## 7. Conclusions

The success of an experimental study depends on the administration of therapeutic agents, and intraspinal drug administration is challenging in mice and rats. Researchers should have adequate knowledge, and studies should be planned and conducted appropriately. Researchers or experimental administrators must consider many factors, and they should have clear knowledge of the anatomy and physiology of the spinal column.

In this review, we described the anatomy of the spinal column; the layers and contents of the spinal meninges; and the location, production, volume, and circulation of CSF in the hopes of drawing careful consideration and attention, contributing to experimental success, and thus minimizing adverse effects in animals. However, further animal research is needed to evaluate the ranges of the epidural, subdural, and intrathecal space thickness in mice and rats based on morphological studies or diagnostic imaging (e.g., micro-computed tomography or ultrasonography). The length of the spinal column, especially from the antero-occipital space to C7, T13, L5, and L6 in mice and rats, should be studied.

This comparative description of injection sites, needle and catheter types and sizes, injection volumes, concentrations, and the purpose and different indications of epidural and intrathecal injections for experimental research in mice and rats will enhance the knowledge in the neuroscience research fields and improve the outcomes of experiments requiring intrathecally and epidurally injected drugs for treating pain, stem cell treatments, and other uses. We concluded that the intrathecal injection route is effective for targeting the DRG in rats and mice. The injection volume is generally larger in epidural injections than in intrathecal injections, possibly because of the consideration of CSF volumes and their pressure. Epidural injection volumes range from 10–50 μL in mice and 10–300 μL in rats. Intrathecal injection volumes range from 5–10 μL in mice and 10–50 μL in rats. Furthermore, the detection of the maximal safe intrathecal and epidural injection volumes in mice and rats is still required. It also has to be made clear whether or not using a different place for each injection while administering multiple doses by acute intrathecal injections would be less painful and harmful than using a single location. The accumulation of information regarding the delivery of drugs through the epidural and intrathecal routes may allow better safety, quality, and reliability in experimental research.

## Figures and Tables

**Figure 1 biomedicines-11-01413-f001:**
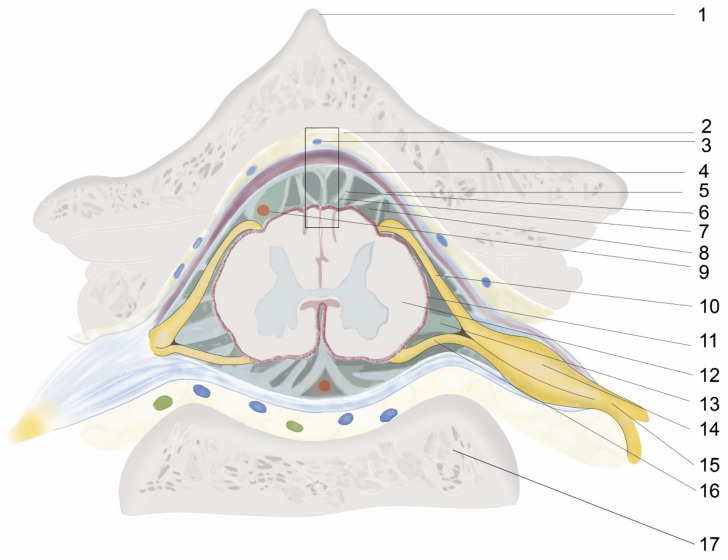
Illustration of cross-sectional anatomy of the spinal colum. 1. Spinous process, 2. Epidural space, 3. Intervertebral venous plexus, 4. Duramater, 5. Arachnoid mater, 6. Arachnoid trabeculae, 7. Figure 2, 8. Subarachnoid space and CSF, 9. Blood vessels, 10. Dorsal nerve root, 11. Pia mater, 12. Spinal cord, 13. Ligamentum denticulatum, 14. Dorsal nerve root ganglion, 15. Dorsal nerve root, 16. Ventral nerve root, 17. Vertebral body.

**Figure 2 biomedicines-11-01413-f002:**
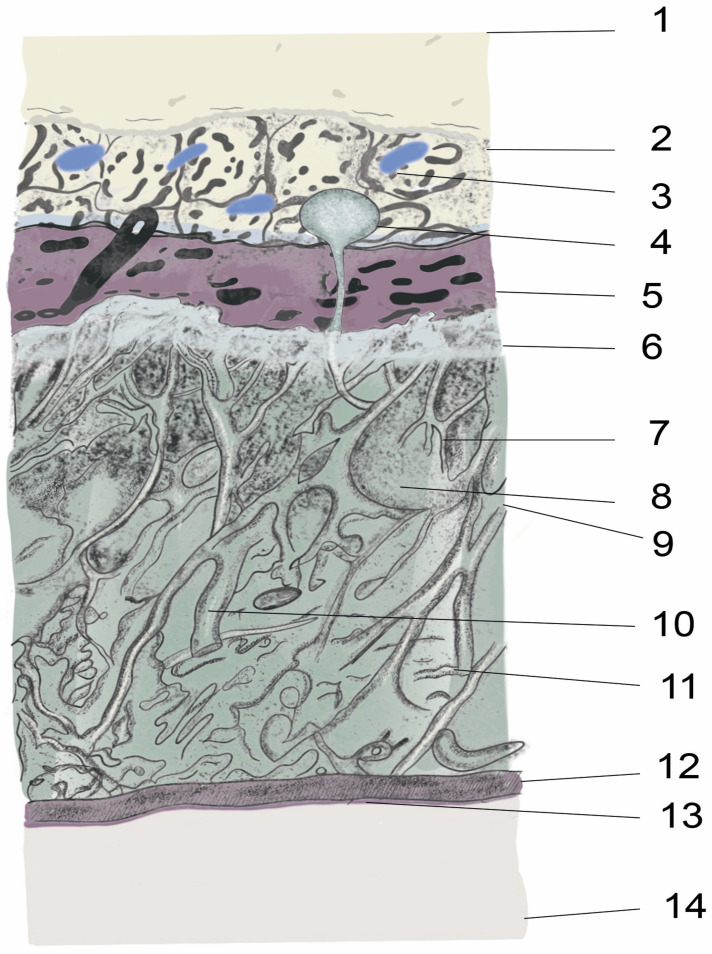
Illustration of anatomy of the spinal meninges and adjacent parts. 1. Ligament flavum, 2. Epidural space, 3. Intervertebral venous plexus, epidural fat 4. Arachnoid villi, 5. Duramater, 6. Arachnoid mater, 7. Arachnoid trabeculae, 8. CSF, 9. Subarachnoid space, 10. Major blood vessel, 11. Collagen fibrils, 12. Pia mater, 13. Glia limitans, 14. Spinal cord.

**Figure 3 biomedicines-11-01413-f003:**
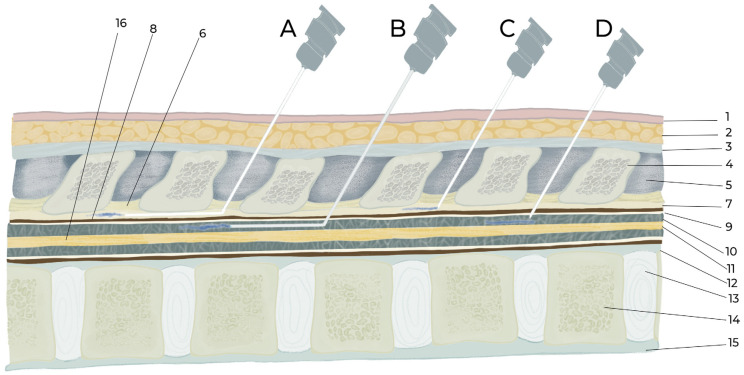
Illustration of intraspinal epidural and intrathecal drug delivery by catheterization and injection. A. Epidural catheterization, B. Intrathecal catheterization, C. Epidural injection, D. Intrathecal injection. 1. Skin, 2. Subcutaneous fat and muscle, 3. Supraspinal ligament, 4. Supraspinal process of vertebra, 5. Interspinous ligament, 6. Ligament flavum, 7. Epidural space, 8. Duramater, 9. Arachnoid mater, 10. Subarachnoid space, 11. Spinal cord, 12. Posterior longitudinal ligament, 13. Inter vertebral disc, 14. Vertebral body, 15. Anterior longitudinal ligament, 16. Spinal cord.

**Figure 4 biomedicines-11-01413-f004:**
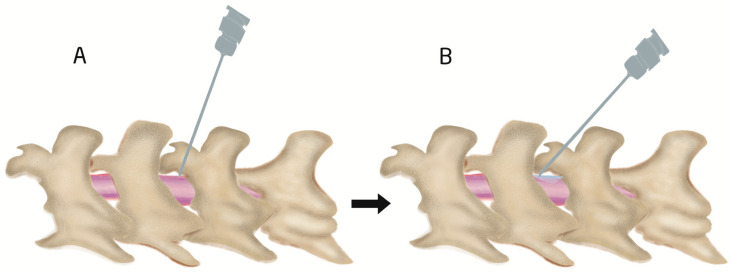
Illustration of intraspinal intrathecal injection site and angle. First insertion of needle at 70–80° angle (**A**) and then reduction until 30–45° (**B**) during drug injection to spread the injected drug easily and to prevent CSF leak out during withdrawing of needle.

**Figure 5 biomedicines-11-01413-f005:**
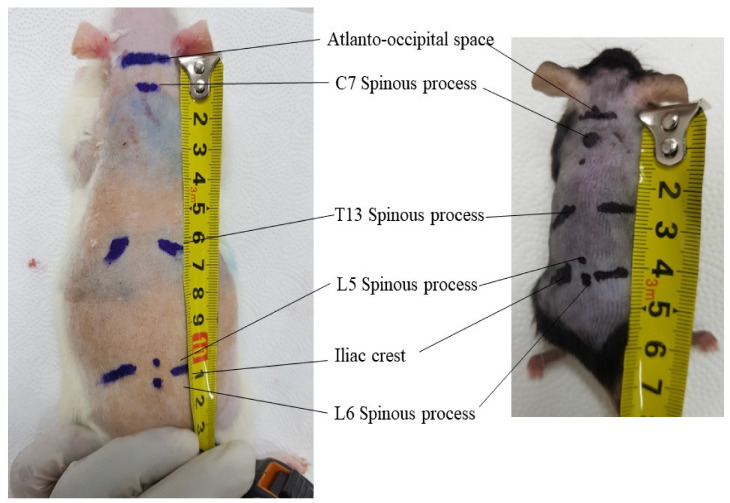
External location of antero-occipital space, C7, T13, L5, L6 spinous process, and iliac crest in 8 weeks old male Sprague Dawley rat and 8 weeks old male C57bl/6 mice. The average distance from antero-occipital space to L6 spinous process is 11 cm in rats and 4.2 cm in mice.

**Table 1 biomedicines-11-01413-t001:** Comparison of epidural and intrathecal space and drug delivery.

	Epidural	Intrathecal
Content	The epidural space contains fat, the dural sac, spinal nerves, blood vessels, and connective tissue.	The subarachnoid space consists of the cerebrospinal fluid (CSF), major blood vessels, and cisterns.
Site of drug delivery	Epidural space.	Intrathecal space, subarachnoid space, or superficial potential space of spinal cord.
Confirmation of proper injection	- No dura mater layer puncture movement of tail, legs - No CSF found in needle.	- Dura mater layer puncture movement of tail, legs present - CSF may be found in needle.
Blood–brain barrier	Mostly avoids absorptive problems made by the blood–brain barrier.	Avoids absorptive problems made by the blood–brain barrier.
Onset of action	After epidural drug administration, drug diffuses through the dura mater into the CSF, which is a significant barrier that needs to be passed, and thus, onset of action is slow.	As the drug is delivered directly into the CSF, the onset of action is fast and instantaneous.
Involvement of systemic circulation	Despite of diffusion of drug through dura mater, some portions of drug also reachss systemic circulation through epidural blood vessels.	There is no involvement of systemic circulation, only restricted within the CSF, circulating in the spinal canal and the brain ventricles.
Dose	Usually 10× that of intrathecal dose, depending on drugs.	Usually 10× lower than epidural dose, depending on drugs.
Occurrence of side effects	Comparatively higher occurrence of side effects due to systemic involvement.	Comparatively lower occurrence of side effects as there is no systemic involvement.
Pain relief	More suitable for short term.	Better for long term.
Uses	Analgesia, anesthesia.	Analgesia, anesthesia, spasticity, chemotherapy, stem cell therapy, antibiotic, protein therapy, etc.

**Table 2 biomedicines-11-01413-t002:** CSF volume, production, and turnover of experimental animals and humans [1].

Species	Volume (μL)	Production (μL/min)	Turnover (Times/24 h)
*Mus musculus*	35–40	0.32–0.35	12–14
*Rattus norvegicus*	150	1.7–2.8	9–12
*Homo sapiens*	100,000–200,000	350–370	3–5

## Data Availability

Not applicable.

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
