# Peer review of "Epidural and Intrathecal Drug Delivery in Rats and Mice for Experimental Research: Fundamental Concepts, Techniques, Precaution, and Application"

_biomedicines, 2023, doi:10.3390/biomedicines11051413_

Round 1

Reviewer 1 Report

this was an interesting paper, well written.

I found the diagrams and tables to be very helpful.

I have a couple of questions:

line 241- when doing multiple seperate injections, do you inject at the same site or a different site? why or why not?

line 256- it ends with rats. puncture  this sentence needs to be improved

What length of needle is used for the intrathecal injection?

line 276 says sterile syringe. Shouldn't this be sterile needle?

line 309 states not to get air bubbles in the line. Why, what effect does this have?

line 359 says confirm proper placement of the catheter. How is this done? if the catheter is in the incorrect position, it is removed and do you try again?

line 408 the word mod-el should be model

line 432 the word col-umn should be column

line 438 the word re-search should be research

line 442 the word stud-ied should be studied

see above for my word corrections

Author Response

Reviewer 1:

Open Review

Quality of English Language

( ) I am not qualified to assess the quality of English in this paper
( ) English very difficult to understand/incomprehensible
( ) Extensive editing of English language required
( ) Moderate editing of English language
(x) Minor editing of English language required
( ) English language fine. No issues detected

Is the work a significant contribution to the field?

Is the work well organized and comprehensively described?

Is the work scientifically sound and not misleading?

Are there appropriate and adequate references to related and previous work?

Is the English used correct and readable?

Comments and Suggestions for Authors

this was an interesting paper, well written.

I found the diagrams and tables to be very helpful.

Thanks for your nice comment. You are highly appreciated by us.

I have a couple of questions:

  • line 241- when doing multiple seperate injections, do you inject at the same site or a different site? why or why not?

Response: Dear reviewer, thank you for raising an important interesting issue. Actually, we have collected the information about multiple intrathecal injections from the published articles which are displayed in the Table 3. There is no information about the drug delivery through the different location. So we have added new sentences to solve this issue-

  1. Epidural and intrathecal injection procedures

Epidural and intrathecal injections can be performed via acute needle puncture or by catheterization. Acute needle puncture is usually used for single drug administration but can also be used for multiple doses [5], and intervals of 24 h, 2 days, and 7 days have been reported. However, multiple injections were performed through the single location e.g., L5-L6 [5, 6], L4-L5 [45] and top of the foramen [46] [Table 3].

Inconclusion: We have added additional sentence -

“It also has to be made clear whether or not using a different place for each injection while administering multiple doses by acute intrathecal injections would be less painful and harmful than using a single location.”.

  • line 256- it ends with rats. puncture this sentence needs to be improved

Response: It was mistake. Now corrected.

“Exact positioning of a needle to the epidural space is difficult noninvasively and re-quires high skill because the distance between the epidural and intrathecal spaces is very small in rats and mice. Therefore, the applications of direct epidural injections using needle puncture in mice and rats are limited because the needle can easily penetrate the dura mater and enter the intrathecal space. Therefore, intrathecal injection by acute needle puncture [5, 6, 17], intrathecal catheterization [11, 13, 14], and epidural catheterization [2-4] but not epidural injection by needle puncture is frequently used in mice and rats”.

  • What length of needle is used for the intrathecal injection?

Response: Dear reviewer,

Size of the needle were listed in the Table 3 (31G, 30 G, 27 G, 25G) but length of the were not described in the published articles.  However, we have described how much length needle should be inserted during injection in rat and mice. Could you please see bellow-

“The depth of needle insertion has been reported to be approximately 0.30 cm in rats after skin and muscle incision and 0.30–0.40 cm without skin and muscle incision in mice [49]”.

“.

  • line 276 says sterile syringe. Shouldn't this be sterile needle?

Response: It was mistake. Now corrected.

“The pelvic girdle is then softly held by one hand to fix the dorsoventral position and, with the other hand, a sterile needle is inserted at the appropriate angle to penetrate the ligamentum flavum and dura mater to reach the arachnoid space".

  • line 309 states not to get air bubbles in the line. Why, what effect does this have?

Response: Dear reviewer, thank you for raising an important interesting issue. We have added the sentences for clarifying this issue with reference-

“However, care should be taken to prevent air bubbles when changing the syringe which may cause adverse effects like alteration of subarachnoid pressure and injury to the nerves or meninges [55]”.

  • line 359 says confirm proper placement of the catheter. How is this done? if the catheter is in the incorrect position, it is removed, and do you try again?

Response: Dear reviewer, we have explained with references how proper placement can be confirmed -

“Appropriate intrathecal positioning can typically be confirmed via the tail-flick test, but this response does not occur every time. Dura mater puncture can also be confirmed by other characteristics such as the formation of an “S” shape by the tail, by hind paw retraction, and occasionally by backflow of the CSF [14,15,20]. After injecting a drug, temporary motor paralysis also occurs, which is a sign of successful drug administration”.

Now we have explained with reference what should be done if failed in first attempt

“For an effective IT injection via acute puncture, only one try is required; however, if the first effort fails, needle should be removed and then second attempt may be made. If the second effort fails, a different intervertebral space should be chosen [15]”.

  • line 408 the word mod-el should be model

Response: Corrected

“Beyond pain management, the intrathecal route is also used for drug administration for the amelioration of spinal injury-induced spasticity [60] and the induction of itching and scratching in behavioral models [78] and a pruritis model [42] (Table 7)”.

  • line 432 the word col-umn should be column

Response: Corrected

“Researchers or experimental administrators must consider many factors, and they should have clear knowledge about the anatomy and physiology of the spinal column”.

  • line 438 the word re-search should be research

Response: Corrected

“However, further animal research is needed to evaluate the ranges of the epidural, subdural, and intrathecal space thickness in mice and rats based on morphological studies or diagnostic imaging (e.g., micro-computed tomography or ultrasonography)”.

  • line 442 the word stud-ied should be studied

Response: Corrected

“The length of spinal column, especially from the antero-occipital space to C7, T13, L5, and L6 in mice and rats, should be studied”.

Comments on the Quality of English Language see above for my word corrections

Submission Date

15 April 2023

Date of this review

24 Apr 2023 20:28:51

Reviewer 2 Report

The authors attempted to describe the intrathecal and epidural routes of administration of substances in laboratory animals. They answered their main question thoroughly. Their statements are supported by the references provided. They organized in tables the scientific evidence of their manuscript argument. Finally, they also refer to the limitations of their article.

Author Response

Reviewer 2:

Open Review

Quality of English Language

( ) I am not qualified to assess the quality of English in this paper
( ) English very difficult to understand/incomprehensible
( ) Extensive editing of English language required
( ) Moderate editing of English language
( ) Minor editing of English language required
(x) English language fine. No issues detected

Is the work a significant contribution to the field?

Is the work well organized and comprehensively described?

Is the work scientifically sound and not misleading?

Are there appropriate and adequate references to related and previous work?

Is the English used correct and readable?

Comments and Suggestions for Authors

The authors attempted to describe the intrathecal and epidural routes of administration of substances in laboratory animals. They answered their main question thoroughly. Their statements are supported by the references provided. They organized in tables the scientific evidence of their manuscript argument. Finally, they also refer to the limitations of their article.

Dear respected Reviewer,

Thanks for your nice comment. You are highly appreciated by us.

Submission Date

15 April 2023

Date of this review

29 Apr 2023 21:09:09
